# Association Between Left Atrial Epicardial Adipose Tissue Attenuation Assessed by Cardiac Computed Tomography and Atrial Fibrillation Recurrence Following Catheter Ablation: A Systematic Review and Meta-Analysis

**DOI:** 10.3390/jcm14134771

**Published:** 2025-07-06

**Authors:** Karol Momot, Kamil Krauz, Michal Pruc, Lukasz Szarpak, Dariusz Rodkiewicz, Artur Mamcarz

**Affiliations:** 1Chair and Department of Experimental and Clinical Physiology, Laboratory of Centre for Preclinical Research, Medical University of Warsaw, Banacha 1b, 02-097 Warsaw, Poland; karol.momot@wum.edu.pl; 23rd Department of Internal Diseases and Cardiology, Międzylesie Specialist Hospital in Warsaw, Medical University of Warsaw, 04-749 Warsaw, Poland; rodkiewicz@gmail.com (D.R.); artur.mamcarz@wum.edu.pl (A.M.); 3Doctoral School, Medical University of Warsaw, 02-097 Warsaw, Poland; 4Department of Clinical Research and Development, LUXMED Group, 02-678 Warsaw, Poland; m.pruc@ptmk.org (M.P.); lukasz.szarpak@gmail.com (L.S.); 5Institute of Biological Science, Collegium Medicum, The John Paul II Catholic University of Lublin, 20-950 Lublin, Poland; 6Henry JN Taub Department of Emergency Medicine, Baylor College of Medicine, Houston, TX 77030, USA

**Keywords:** epicardial adipose tissue, computed tomography, atrial fibrillation, catheter ablation, pulmonary vein isolation

## Abstract

**Background:** Epicardial adipose tissue (EAT) may contribute to the pathogenesis of atrial fibrillation (AF). The attenuation of EAT assessed by means of computed tomography (CT) may reflect the severity of local tissue inflammation. This study aimed to systematically analyze the relationship between the attenuation of EAT surrounding the left atrium (LA-EAT) and AF recurrence after catheter ablation (CA). **Methods:** Five databases were searched up to April 10, 2025. Original studies involving adult patients with paroxysmal or persistent AF undergoing CA were included if they provided quantitative measurements of LA-EAT attenuation on cardiac CT before ablation. **Results:** A total of seven retrospective observational studies with 2440 patients were included in the analysis. The percentage of male participants ranged from 60.9% to 73.0%, and the mean or median age of patients varied from 57.5 to 68.2 years. The mean body mass index across studies ranged from 24.0 to 32.4 kg/m^2^. A pooled analysis of all included trials demonstrated that the mean LA-EAT attenuation in the AF recurrence group was −78.97 ± 15.34 HU, which was less negative compared to the non-recurrence group (−81.37 ± 15.46 HU; mean difference [MD] = 2.22; 95% confidence interval [CI]: 0.84 to 3.61; *p* = 0.002). **Conclusions:** LA-EAT attenuation is significantly more positive in patients experiencing AF recurrence compared to those without recurrence following CA procedures.

## 1. Introduction

Atrial fibrillation (AF) is the most common sustained arrhythmia encountered in clinical practice [1], with an estimated lifetime risk affecting approximately one in every three to five individuals after the age of 45. Globally, between 2010 and 2019, the prevalence of AF increased significantly, from about 33.5 million to 59 million affected patients [2]. Catheter ablation (CA) is widely used as a primary treatment method for AF [3]. However, the recurrence rates after this procedure remain notably high, with only about 50% of patients remaining free from arrhythmia five years after a single CA procedure and approximately 30% of patients after multiple CA procedures [4]. Thus, the identification of novel, reliable predictors of AF recurrence, complementing known clinical variables such as age, body mass index, arrhythmia duration, or comorbid conditions like chronic obstructive pulmonary disease and obstructive sleep apnea, may substantially improve patient selection and ablation outcomes.

Epicardial adipose tissue (EAT), easily visualized by pre-procedural computed tomography (CT), has gained significance in the context of cardiovascular disease pathogenesis [5]. EAT is the visceral fat depot around the heart that may interact locally with the myocardium [6]. It could exert a protective effect through the secretion of adiponectin and adrenomedullin in response to local or systemic metabolic or mechanical stress [7]. Conversely, EAT may also adversely affect the myocardium via the paracrine release of pro-inflammatory and pro-fibrotic cytokines [8], thus facilitating myocardial fibrosis, a significant substrate for AF. Furthermore, substances released by EAT may affect cardiomyocyte electrophysiology by altering ion currents and electrical coupling, potentially leading to arrhythmogenesis [9].

Multiple studies have investigated the relationship between the EAT volume and AF recurrence after CA, and their findings have been summarized in three meta-analyses [10,11,12]. Two of these analyses suggest that the total EAT volume or thickness is associated with AF recurrence in patients undergoing CA. However, the third analysis showed that only the EAT volume specifically surrounding the left atrium (LA), rather than the total EAT volume, significantly predicts AF recurrence following CA. The results of this recent meta-analysis indicate that the pathophysiological function of EAT may differ according to its location.

However, it is essential to note that the EAT volume is not the only parameter assessable by pre-procedural cardiac CT. A relatively novel approach involves the measurement of EAT attenuation, a method adapted from assessments of perivascular adipose tissue (PVAT) inflammation [13,14]. Previous studies suggest that EAT attenuation measured by cardiac CT may reflect the severity of local inflammation, correlating with findings from histological examinations and positron emission tomography (PET) [14]. It is established that inflammation shifts attenuation toward higher (more positive) values [15]. Given the close anatomical relationship between LA-EAT and the LA, specifically assessing LA-EAT attenuation could represent a promising imaging approach to identifying inflammation in this region. This fat depot lies in direct contact with the atrial myocardium targeted during CA, and its inflammatory profile may influence lesion durability and ablation efficacy by modulating tissue resistivity, edema formation, or healing responses. Moreover, measuring EAT attenuation appears to be more reliable and accurate compared to volume assessments, as attenuation values demonstrate significantly less variability compared to volume measurements, which strongly depend on subjective operator decisions, such as selecting the number of slices to analyze and defining the initial and final slices [16,17]. Consequently, measurements of EAT attenuation seem to demonstrate notably higher reproducibility compared to EAT volume assessments. However, it is essential to consider that common comorbidities associated with systemic inflammation, such as obesity or aging, may influence the phenotype of EAT to varying degrees, just as they differentially affect the risk and course of AF itself. The underlying hypothesis is that EAT may not merely reflect systemic risk but act as a direct effector contributing to atrial arrhythmogenesis by mediating inflammatory and fibrotic processes affecting the adjacent LA myocardium.

This study aims to enhance the existing knowledge by systematically evaluating the link between LA-EAT attenuation and AF recurrence after CA.

## 2. Methods

This meta-analysis was conducted on the basis of the Preferred Reporting Items for Systematic Reviews and Meta-Analyses (PRISMA) statement (Appendix A) [18]. We gave priority to registering the analysis plan on the International Prospective Register of Systematic Reviews (PROSPERO) before we proceeded (record CRD420251025994).

### 2.1. Search Strategy

A comprehensive literature search was performed using five electronic databases: PubMed, Embase, Cochrane Library, Web of Science, and CINAHL. Searches covered all records from their inception to 10 April 2025.

The search strategy incorporated both free-text keywords and controlled vocabulary terms (e.g., MeSH in PubMed and Emtree in Embase) related to “atrial fibrillation”, “epicardial adipose tissue”, “computed tomography”, “attenuation”, and “catheter ablation”. Boolean operators (AND, OR) were applied to optimize the sensitivity and specificity of the search. The detailed search syntax for each database is available in Appendix A.

No limitations were imposed regarding the publication date or study design. Only studies published in English were deemed eligible. To guarantee thorough literature coverage, the reference lists of all included articles and pertinent systematic reviews were manually examined for additional research not discovered during the original database search. Furthermore, backward citation tracking was conducted via Google Scholar to uncover possibly pertinent studies that might have been overlooked.

### 2.2. Inclusion and Exclusion Criteria

Studies were selected based on predefined inclusion and exclusion criteria.

Inclusion Criteria: Original studies (prospective or retrospective) involving adult patients with paroxysmal or persistent AF undergoing CA (radiofrequency or cryoballoon ablation). Studies were required to provide quantitative measurements of LA-EAT attenuation on cardiac CT before ablation, employing established Hounsfield unit (HU) ranges. Included studies were required to provide AF recurrence outcomes throughout a follow-up period of no less than 6 months post-ablation, using a 3-month post-ablation blanking period for the definition of AF recurrence.Exclusion Criteria: We eliminated case reports, conference abstracts, review papers, editorials, and research that lacked original patient data. We eliminated studies that did not specifically evaluate LA-EAT attenuation, such as those that simply assessed the EAT volume or thickness without attenuation data. If many publications presented overlapping patient groups, the most complete or latest study was selected to prevent data duplication. We also eliminated research in languages other than English or without a clearly defined AF recurrence outcome post-ablation.

### 2.3. Study Selection

The studies obtained from the literature search were imported to the EndNote X9 software (Clarivate Analytics, Philadelphia, PA, USA), and duplicate records were removed. Two authors (K.M. and M.P.) independently reviewed the studies based on their titles and abstracts. This was followed by a review of the full texts of the articles. A third author (L.S.) was consulted in the event of any disagreements. The study selection process was documented in a PRISMA flow diagram illustrating the number of studies identified, screened, excluded, and ultimately included in the meta-analysis (Figure 1).

### 2.4. Data Extraction

Data from each included study were extracted by two investigators (M.P. and K.M.) independently using a standardized data extraction form. Disagreements were resolved through consulting with the senior author (L.S.). The extracted data were entered by K.M. and M.P. and verified by L.S. and A.M. Extracted data included the study characteristics (first author, publication year, country, study design), patient demographics (sample size, mean age, sex distribution, body mass index), AF type (paroxysmal vs. persistent), EAT assessment methods, ablation details (energy source and technique used), CA success rate, and follow-up duration.

We extracted data from the included studies using a standardized, predesigned form and then entered them into a Microsoft Excel spreadsheet (Microsoft Corporation, Redmond, WA, USA). In instances of missing or incomplete information regarding primary outcomes, we intended to reach out to the corresponding authors of the relevant studies to acquire supplementary data.

### 2.5. Risk of Bias Assessment

The risk of bias for each included study was evaluated using the Newcastle–Ottawa Scale (NOS), a recognized instrument for assessing the methodological quality of observational studies [19]. Each study was independently evaluated by two reviewers across the three primary domains of the Newcastle–Ottawa Scale: the selection of study participants, the comparability of study groups, and the ascertainment of outcomes. Each study received a score of 0 to 9 stars according to established criteria. Studies that received scores of 7–9 stars were classified as having high quality, reflecting a low risk of bias. Those with scores of 4–6 stars were categorized as having a moderate risk, while scores below 4 indicated a high risk of bias. Discrepancies in scoring among reviewers were addressed through discussion and consensus. The NOS scores, while not serving as an exclusion criterion, informed subsequent sensitivity analyses and the interpretation of the overall study findings.

### 2.6. Data Synthesis and Analysis

Statistical analyses were performed using the Review Manager software (version 5.4, Nordic Cochrane Centre, Cochrane Collaboration) and Stata (version 14, StataCorp, College Station, TX, USA). Odds ratios (ORs) with 95% confidence intervals (CIs) were used as the measures of effect for dichotomous outcomes, whereas mean differences (MDs) with 95% CIs were used for continuous outcomes. In cases where a study reported a continuous outcome as a median with a range and interquartile range (IQR) rather than a mean with a standard deviation (SD), we estimated the mean (SD) for that study using the method described by Hozo et al. [20]. A random-effects model (DerSimonian–Laird method) was employed to account for the expected between-study variability in measurements and patient populations. We chose a random-effects model a priori. Statistical heterogeneity was assessed with the Cochran’s Q chi-squared test and quantified by the I^2^ statistic, considering I^2^ < 25% to indicate low heterogeneity, 25–50% to indicate moderate heterogeneity, and >50% to indicate high heterogeneity [21]. To evaluate the robustness of the pooled estimates, we conducted sensitivity analyses by sequentially omitting each individual study (leave-one-out method). To find out if certain study-level covariates contributed to the variation in the effect size of the relationship between AF recurrence and LA-EAT attenuation, a meta-regression analysis was conducted. The mean difference in LA-EAT attenuation between patients with and without AF recurrence served as the dependent variable. Covariates such as the mean age, body mass index (BMI), percentage of patients with paroxysmal AF, length of follow-up (in months), and percentage of patients treated with radiofrequency ablation (RFA) were assessed. In order to account for the accuracy of each study’s effect estimate, inverse variance weighting was applied to a weighted least squares (WLS) regression model. A two-tailed *p*-value < 0.05 was considered statistically significant.

## 3. Results

### 3.1. Description of Selected Studies

A total of 1874 articles were retrieved through searching seven databases. All articles were exported to EndNote, with 1408 duplicates removed, leaving the remaining articles for title and abstract review. Based on the inclusion/exclusion criteria, 26 articles were selected for full-text review. Ultimately, seven articles were included in the final analysis. The detailed screening process and reasons for exclusion are shown in Figure 1.

Seven retrospective observational studies fulfilled the established inclusion criteria and were incorporated into this meta-analysis [15,22,23,24,25,26,27]. The research was performed in several geographical regions, including Europe (Austria, France, Hungary), Asia (China, Japan), and North America (USA), and was published from 2019 to 2024. All selected studies examined the correlation between LA-EAT attenuation, as evaluated by cardiac CT, and the recurrence of AF subsequent to CA. None of the studies included patients undergoing repeat ablation procedures or those with longstanding persistent AF, with the exception of one study by Yang et al., which did not clearly specify whether the ablation procedures were exclusively first-time interventions [26].

The studies exhibited variability in their sample sizes, spanning from 43 to 732 participants, and follow-up periods, which ranged from 11.5 to 31 months. All investigations utilized cardiac CT to evaluate LA-EAT attenuation, but with varied Hounsfield unit (HU) ranges and tube voltage settings. Notwithstanding methodological discrepancies, each trial yielded adequate outcome data for quantitative synthesis.

### 3.2. Study Characteristics

Table 1 summarizes the baseline characteristics of the included studies. Additional details regarding the technical aspects of CT imaging are presented in Appendix A. Across the seven studies, the researchers evaluated a total of 2440 patients with AF undergoing CA. The proportion of male participants ranged from 60.9% to 73.0%, and the mean or median age of patients varied from 57.5 to 68.2 years. The mean BMI across studies ranged from 24.0 to 32.4 kg/m^2^.

All studies utilized contrast-enhanced cardiac CT imaging to assess EAT attenuation specifically in the region surrounding the LA. Most studies employed three-dimensional segmentation techniques, except Ciuffo et al. [27], who used a two-dimensional approach in standard chamber views. The HU ranges defining LA-EAT attenuation varied slightly, from −250 to −5 HU, with the differences likely influenced by the scanner settings and image acquisition parameters (e.g., tube voltage and reconstruction algorithms).

All studies included patients undergoing CA with energy sources, including RFA and cryoballoon ablation (CBA). The use of RFA vs. CBA differed among centers, with some studies reporting a mixed-modality approach.

All studies demonstrated high methodological quality, with NOS scores ranging from 7 to 8 out of 9 points (Table 1).

### 3.3. Results from Meta-Analysis

The pooled analysis of all included trials demonstrated that the mean LA-EAT attenuation in the AF recurrence group was −78.97 ± 15.34 HU, which was significantly higher (less negative) compared to the non-recurrence group (−81.37 ± 15.46 HU; mean difference [MD] = 2.22; 95% confidence interval [CI]: 0.84 to 3.61; *p* = 0.002) (Figure 2).

In a subgroup analysis limited to studies utilizing three-dimensional (3D) imaging for LA-EAT quantification, the attenuation values also differed significantly between the recurrence and non-recurrence groups, amounting to −73.59 ± 13.55 HU and −76.44 ± 13.81 HU, respectively (MD = 2.20; 95% CI: 0.42 to 3.98; *p* = 0.02).

To mitigate potential confounding variables, we performed an exploratory weighted meta-regression incorporating study-level covariates: the mean age, BMI, the proportion of patients with paroxysmal AF, the follow-up duration, and the proportion of patients undergoing RFA. None of these variables substantially influenced the observed correlation between LA-EAT attenuation and AF recurrence (Table 2). These possible factors do not entirely account for the association between elevated (less negative) LA-EAT attenuation and AF recurrence, according to the findings.

## 4. Discussion

To our knowledge, this is the first meta-analysis to evaluate LA-EAT attenuation, specifically in the context of AF ablation outcomes. By pooling data from seven retrospective observational studies, including 2440 AF patients, our analysis demonstrated that pre-procedural LA-EAT attenuation measured by cardiac CT was more positive in patients who experienced AF recurrence after CA compared to those without recurrence.

The link between inflamed LA-EAT and AF recurrence after catheter ablation can be explained by at least two interrelated pathophysiological mechanisms. First, increased local inflammation—as reflected by less negative LA-EAT attenuation values—may impair adequate lesion formation during ablation by promoting atrial wall edema and increasing tissue resistivity, thereby reducing lesion transmurality and enhancing the likelihood of pulmonary vein reconnection [28]. Second, beyond its acute effects on lesion formation, LA-EAT may chronically contribute to atrial remodeling through the secretion of pro-inflammatory and pro-fibrotic mediators, which promote the development of atrial fibrosis [29,30]. Previous studies have reported an association between LA-EAT attenuation and the presence of low-voltage zones in the left atrium, which are commonly regarded as a surrogate for atrial fibrosis in patients with atrial fibrillation [31]. The presence of fibrosis facilitates intra-atrial re-entry circuits and may lead to arrhythmia recurrence even in patients with durable pulmonary vein isolation [28]. Given its proximity to the LA and pro-inflammatory potential, LA-EAT attenuation may serve as a tissue-specific biomarker relevant to both acute lesion formation and long-term ablation success, with the advantage of being easily accessible through routine pre-procedural cardiac CT.

LA-EAT assessment may not only provide prognostic insights into arrhythmia recurrence following catheter ablation but could also potentially be relevant in the broader context of heart failure prediction. As highlighted in recent studies, LA dilation has been associated with adverse outcomes in patients with heart failure and dilated cardiomyopathy [32]. Whether local atrial inflammation—as captured by LA-EAT attenuation—contributes to this prognostic dimension remains speculative but warrants further investigation. Supporting this hypothesis, Zhao et al. [33] demonstrated that, even in individuals without AF, an increased EAT volume was independently associated with subclinical LA dysfunction, reflected by impaired reservoir function and reduced longitudinal strain. These findings suggest that EAT may have mechanical effects on the atrial myocardium before overt structural or electrical disease becomes evident.

The modification of LA-EAT parameters through pharmacological and lifestyle interventions has attracted increasing interest. Several anti-diabetic agents, particularly GLP-1 receptor agonists and SGLT2 inhibitors, have been shown to reduce the EAT volume and modulate its inflammatory profile [34,35,36]. Lifestyle changes such as regular exercise and weight loss are also associated with decreased EAT volumes [37,38], and, notably, many of these interventions have been independently linked to improved PVI outcomes and reduced AF recurrence. A meta-regression found that each 1% absolute reduction in body weight translated into a 6% relative reduction in the risk of AF recurrence after ablation (RR = 0.94 [0.90–0.98]), regardless of the weight loss method used [39]. Regarding pharmacotherapy, a recent meta-analysis showed that GLP-1 receptor agonists were associated with reduced AF recurrence in patients undergoing ablation [40]. Additionally, a meta-analysis demonstrated that SGLT2 inhibitor use after ablation was associated with a 39% reduction in AF recurrence (RR = 0.61 [0.49–0.77]) [41].

Nonetheless, none of these studies directly evaluated whether the observed benefits were mediated through the modulation of EAT, and no causal relationship between changes in LA-EAT attenuation and reduced AF recurrence has been established to date, underscoring the need for mechanistic prospective research.

Notably, two studies in our meta-analysis adopted distinct methodologies for evaluating LA-EAT attenuation. Ciuffo et al. [27] employed a two-dimensional approach by manually segmenting LA-EAT in standard four-chamber and two-chamber views, rather than the conventional three-dimensional analysis. The mean attenuation of LA-EAT assessed in the four-chamber view significantly differed between the recurrence and control groups (−92.0 ± 9.8 vs. −96.5 ± 9.4, *p* = 0.006). In contrast, no difference was observed regarding two-chamber LA-EAT attenuation (−79.7 ± 10.3 vs. −81.5 ± 11.3, *p* = 0.328). Mahdiui et al. [15] performed a three-dimensional analysis but focused exclusively on the EAT posterior to the LA. No differences were observed between the recurrence and non-recurrence groups (−95.6 ± 6.2 vs. −96.8 ± 8.7, *p* = 0.098). However, the number of patients with posterior LA adipose tissue attenuation ≥ −96.4 HU was significantly higher in the recurrence group (100 (60%) vs. 145 (50%), *p* = 0.041). These results suggest that LA-EAT attenuation should be assessed more globally rather than focusing on specific slices or regions, as limited or localized measurements, although easier to perform, may lead to biased or inconsistent results. After including these studies in a separate subgroup, we found that these alternative methods did not yield statistically significant differences in LA-EAT attenuation between the recurrence and non-recurrence groups (*p* = 0.12). Consequently, future studies should prioritize standardized three-dimensional LA-EAT attenuation measurements to enhance the consistency and accuracy.

Several studies investigating the general association between EAT attenuation and AF recurrence were excluded from our meta-analysis, as they did not directly compare the mean attenuation values between the recurrence and non-recurrence groups. Cruz et al. [42] divided patients into two groups based on EAT attenuation: those with attenuation ≥−75 HU and those with attenuation <−75 HU. The analysis showed that these groups did not differ significantly in terms of survival free from AF recurrence. Moreover, EAT attenuation in this study was assessed from a single slice located posterior to the LA, rather than using a three-dimensional analysis. Huber et al. (2022) demonstrated that, in a univariate Cox regression analysis for AF recurrence after 1 year, LA-EAT attenuation was associated with a hazard ratio of 1.6 (95% confidence interval: 1.0–2.5; *p* = 0.06) [43]. Similar results were reported by the same research group in a more recent study published in 2024 [44]. Additionally, Sang et al. [45] analyzed the impact of LA-EAT attenuation on AF recurrence after CA in patients with both paroxysmal and persistent AF. Univariate Cox survival analysis revealed that, in both subgroups, LA-EAT attenuation was associated with AF recurrence. Feng et al. [46], in a study published in 2025, extensively investigated EAT in the context of AF ablation outcomes. However, their analysis was limited to total EAT and periatrial EAT, without a specific assessment of LA-EAT, which precluded the inclusion of their data in this meta-analysis. Among all the analyzed parameters of EAT, including the periatrial and total EAT attenuation and volume, only the total EAT volume was identified as a significant predictor of AF recurrence after CA. This finding further supports our hypothesis that the regional evaluation of LA-EAT, rather than global or periatrial fat metrics, may offer greater pathophysiological relevance and predictive value in the context of catheter ablation outcomes.

An important methodological consideration when interpreting studies on EAT attenuation is the potential variability in attenuation values arising from differences in the tube voltage used during image acquisition. For instance, attenuation values obtained from scans performed at 100 kV should ideally be adjusted using a validated conversion factor of 1.11485 to correspond with measurements derived from 120 kV scans [14,17]. Although this adjustment was not consistently applied across the analyzed studies, it reflects an evolving aspect of the EAT quantification methodology. Additionally, it is worth noting that the attenuation ranges used to define EAT varied between studies, from −250/−195 HU to −50/−5 HU. These differences likely contributed to some variability in the reported mean attenuation values. On the other hand, previous validation studies—although conducted on perivascular adipose tissue (PVAT)—have shown that technical (e.g., tube voltage, lumen attenuation) and anatomical (e.g., vessel diameter) parameters are only weakly associated with PVAT attenuation, influencing its values to a minimal extent [14].

While our meta-analysis provides insights into the association between LA-EAT attenuation and AF recurrence, this parameter remains at an early stage of investigation. The growing interest in adipose tissue imaging highlights its potential relevance, but clinical applicability will require further prospective studies and standardization efforts. The harmonization of CT acquisition protocols and attenuation thresholds across centers, supported by expert consensus and multicenter collaboration, may help to define the role of LA-EAT attenuation in future personalized risk stratification strategies for AF patients.

### Limitations of the Study

Our meta-analysis has several limitations. Firstly, all included studies were retrospective, potentially introducing selection and reporting biases compared with prospective cohort studies. However, given that this is a relatively novel research area, no prospective studies evaluating the association between LA-EAT attenuation and AF recurrence have been published to date. Additionally, there was notable variability in the reported LA-EAT attenuation values across different study centers, with the mean LA-EAT attenuation for the recurrence group ranging from −93.7 to −61.4 HU and for the non-recurrence group from −93.4 to −62.8 HU. This variability likely reflects differences in CT acquisition protocols, including the tube voltage settings and attenuation ranges applied for EAT quantification. Given this heterogeneity in imaging protocols, the current evidence does not allow for the determination of a standardized attenuation threshold that would carry consistent clinical implications. However, within individual centers, attenuation measurements were more internally consistent, and all studies demonstrated clear and statistically significant differences between patients with and without AF recurrence. While we acknowledge the potential for residual confounding, our exploratory meta-regression did not identify age, BMI, the AF type, the follow-up duration, or the ablation technique as significant moderators of the association between LA-EAT attenuation and AF recurrence. Given the small number of included studies (n = 7), we recognize the limited statistical power of this analysis and recommend interpreting these results with appropriate caution. Nonetheless, this finding supports the internal consistency of our pooled effect estimate and provides preliminary evidence that LA-EAT attenuation may function as a recurrence-related imaging biomarker, independently of major clinical variables.

Despite these limitations, this meta-analysis, based on a large, pooled sample, provides supportive and preliminary evidence for the association between LA-EAT attenuation and AF recurrence after CA. While our analysis does not establish a causal relationship or confirm LA-EAT attenuation as an independent predictor, the consistency of the findings within individual studies suggests that this parameter may hold translational potential. We propose it as a complementary—not stand-alone—biomarker that could enhance the current risk stratification strategies when interpreted alongside clinical and imaging data. Prospective studies are warranted to evaluate its independent prognostic value and define its role in personalized ablation planning.

## 5. Conclusions

The present meta-analysis indicates that LA-EAT attenuation is significantly more positive in patients experiencing AF recurrence compared to those without recurrence following CA procedures. LA-EAT attenuation measurement may represent a promising imaging tool for future use in pre-procedural risk assessment. While current data support an association with arrhythmia recurrence, further prospective, confounder-adjusted studies are required before its integration into clinical decision-making or routine imaging protocols.

## Figures and Tables

**Figure 1 jcm-14-04771-f001:**
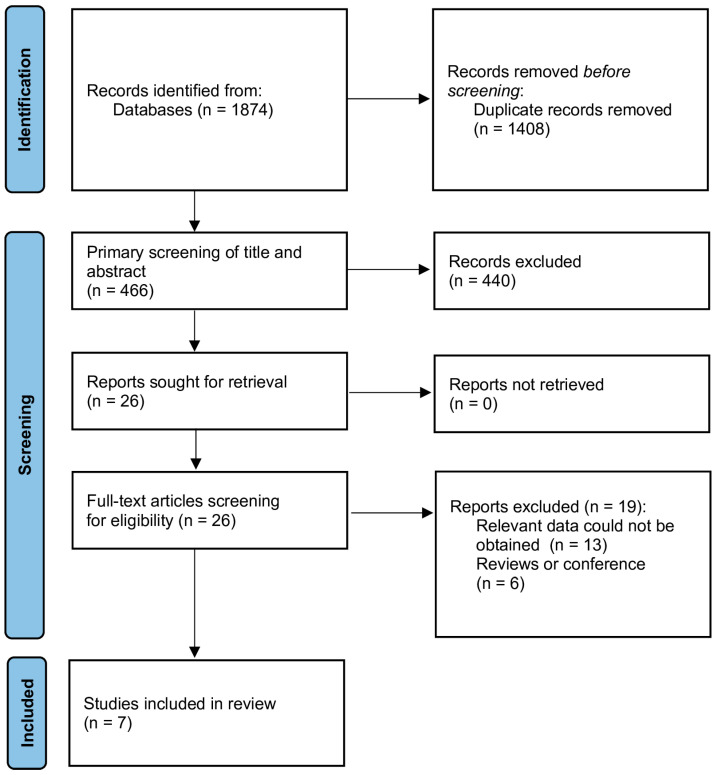
PRISMA flowchart.

**Figure 2 jcm-14-04771-f002:**
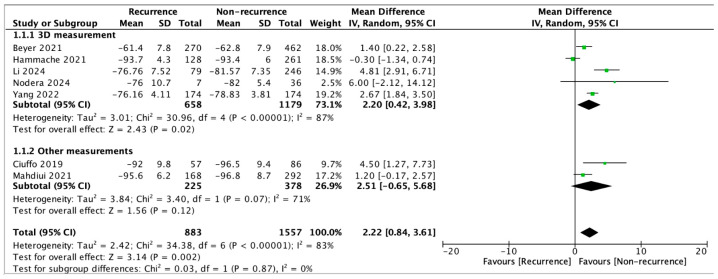
Forest plot of pooled left atrial epicardial adipose tissue (LA-EAT) attenuation between patients with and without atrial fibrillation (AF) recurrence after ablation procedure [15,22,23,24,25,26,27].

**Table 1 jcm-14-04771-t001:** Summary of the included studies.

Author, Year	Region	Study Type	N	Males (%)	Age (y)	BMI	Paroxysmal AF (%)	Follow-Up (m)	EAT Assessment Methods	CA Method	Additional Ablation Lesions	CA Success Rate (%)	NOS Score
Beyer, 2021 [22]	Innsbruck, Austria	Retrospective, OA	732	73%	57.5	26.9	88.4%	31 m	3D; EAT surrounding LA	RFA 36.9% CBA 63.1%	0%	63.1%	8
Hammache, 2021 [23]	Vandœuvre-lès-Nancy, France	Retrospective, OA	389	65.8%	58.1	27.1	100%	12 m	3D; EAT surrounding LA	RFA 100%	0%	67.1%	7
Li, 2024 [24]	Xuzhou, China	Retrospective, OA	325	60.9%	60.6	25.5	59.1%	11.5 m	3D; EAT surrounding LA	RFA 100%	No data	75.7%	8
Nodera, 2024 [25]	Fukui, Japan	Retrospective, OA	43	65%	68.2	24.0	53%	20 m	3D; EAT surrounding LA	RFA 20.9% CBA 79.1%	25.6%	83.8%	7
Yang, 2022 [26]	Shanghai, China	Retrospective, OA	348	63.5%	Non-recurrence: 63.0; Recurrence: 64.0 (medians)	Non-recurrence: 24.49; Recurrence: 24.68 (medians)	100%	12 m	3D; EAT surrounding LA	CBA 100%	No data	74.4%	8
Ciuffo, 2019 [27]	Baltimore, USA	Retrospective, OA	143	63.6%	62.2	32.4	59.4%	at least 12 m	2D; EAT surrounding LA	RFA 74.1% CBA 25.9%	11.9%	60.1%	8
Mahdiui, 2021 [15]	Budapest, Hungary	Retrospective, OA	460	65.7%	61	29	77.0%	18 m (median)	3D; EAT posterior to LA	RFA 100%	No data	63.5%	8

Abbreviations: 2D, two-dimensional; 3D, three-dimensional; AF, atrial fibrillation; BMI, body mass index; CA, catheter ablation; CBA, cryoballoon ablation; CT, computed tomography; EAT, epicardial adipose tissue; HU, Hounsfield unit; LA, left atrium; m, months; OA, original article; PVI, pulmonary vein isolation; RFA radiofrequency ablation; y, years.

**Table 2 jcm-14-04771-t002:** Summary of weighted meta-regression results assessing potential confounders affecting the mean difference in LA-EAT attenuation between patients with and without AF recurrence.

Covariate	Coefficient (β)	*p*-Value	95% CI Lower	95% CI Upper
Intercept	14.51	0.630	−266.16	295.19
Age	−0.23	0.551	−3.61	3.16
BMI	0.27	0.391	−2.16	2.70
% Paroxysmal AF	−0.03	0.534	−0.51	0.44
Follow-up (months)	−0.09	0.439	−1.09	0.90

Abbreviations: AF, atrial fibrillation; BMI, body mass index.

## Data Availability

The data that support the findings of this study are available from the corresponding author upon reasonable request.

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
