# Peer review of "Association Between Left Atrial Epicardial Adipose Tissue Attenuation Assessed by Cardiac Computed Tomography and Atrial Fibrillation Recurrence Following Catheter Ablation: A Systematic Review and Meta-Analysis"

_jcm, 2025, doi:10.3390/jcm14134771_

Round 1
Reviewer 1 Report
Comments and Suggestions for Authors
A nice meta analysis of studies evaluating LA EAT
attenuation and post ablation AF recurrence showing
A robust and significanr association between these
two.
The introdiction and methid are clearly written.
Few comments:
- Table 1 - nor clear why 'redo' ablation written as mostIf these were first ablatiobs.
- Also in figure 2 the legend speaks of EAT volumeand should be changed to EAT attenuation.
2.. The discusion needs revision. The authors speak
Highly of LA EAT as tgerapuetic target bur no causuality
was shown but rather an association
Accordingly, further studies needed to show EAT attenuation change to
cause difference in AF recurrence.
3. Lastly, the discussion on the impact of imaging protocol
on EAT attenuation is important and nicely done.
Reviewer 2 Report
Comments and Suggestions for Authors
This is a well-executed and timely meta-analysis investigating the association between left atrial epicardial adipose tissue (LA-EAT) attenuation and atrial fibrillation (AF) recurrence after catheter ablation. The methods are sound, and the analysis is robust. However, a few aspects should be addressed to enhance the manuscript's clinical relevance and scientific impact.
Major Comments
1) The authors describe the mechanistic role of EAT inflammation but should integrate these mechanisms more thoroughly into the interpretation of the results. Specifically, how might the shift in less negative HU values affect ablation outcomes? A brief synthesis of the clinical-pathophysiological link would help translate findings into practice.
2) A relevant aspect is the emerging evidence linking the radiodensity of LA epicardial adipose tissue to the electrophysiological properties of the atrial myocardium. This relationship may provide an additional pathophysiological explanation for the findings reported in this meta-analysis and should be briefly discussed (10.1007/s00330-018-5793-4).
3) The role of longitudinal assessment of the left atrium, independently from left ventricular function trends, has also emerged as crucial in the prognostication of patients with heart failure. Whether local inflammation of the left atrium—assessed by adipose tissue attenuation—could play a role in this scenario should be at least speculatively discussed (10.1016/j.echo.2022.10.017).
4) While the study acknowledges differences in attenuation ranges and CT acquisition parameters across studies, it does not clearly address how these differences may limit clinical applicability. The authors should suggest concrete steps or consensus statements for future standardization of LA-EAT attenuation measurements across centers and devices.
Minor Comments
1) Overall, the manuscript is clearly written, but minor grammatical corrections and smoother phrasing are needed in some sections (e.g., abstract and limitations). A light language revision is recommended.
2) Table 1 is rich in data but quite dense. Consider simplifying by removing redundant elements (e.g., repeated phrasing) or using footnotes to define terms like PVI, RFA, and CBA.
3) The manuscript mentions five databases but does not show the full search strings in the main text or an appendix. Including at least one representative search string (e.g., from PubMed) would improve reproducibility.
4) Ensure consistent terminology throughout the manuscript. For instance, “recurrence group” and “with recurrence group” should be unified. Similarly, consider always referring to “left atrial epicardial adipose tissue” as “LA-EAT” after first definition.
Round 2
Reviewer 1 Report
Comments and Suggestions for Authors
No further comments
Author Response
Thank you for your time and effort
Reviewer 2 Report
Comments and Suggestions for Authors
Thank you for addressing my comments. I have no further suggestions.
Author Response
Thank you for your time and effort